# Near-Infrared (NIR) Spectroscopy as an Alternative for Predicting n-Alkane Concentration in Excreta of Laying Hens: NIR-Generated Data for Dietary Composition Estimation

**DOI:** 10.3390/ani14050806

**Published:** 2024-03-05

**Authors:** Laid Dardabou, José Carlos Martínez Ávila, Markus Werner Schmidt, Károly Dublecz, Christiane Schwarz, Miguel Angel Ibáñez, Martin Gierus

**Affiliations:** 1Institute of Animal Nutrition, Livestock Products, and Nutrition Physiology, University of Natural Resources and Life Sciences, 1190 Vienna, Austria; laid.dardabou@boku.ac.at (L.D.); markus.schmidt@boku.ac.at (M.W.S.); christiane.schwarz@boku.ac.at (C.S.); 2Departamento de Economía Agraria, Estadística y Gestión de Empresas, Universidad Politécnica de Madrid, 28040 Madrid, Spain; jc.martinez.avila@upm.es (J.C.M.Á.); miguel.ibanez@upm.es (M.A.I.); 3Institute of Physiology and Nutrition, Georgikon Campus, Hungarian University of Agriculture and Life Sciences, 8360 Keszthely, Hungary; dublecz.karoly@uni-mate.hu

**Keywords:** near-infrared spectroscopy, n-alkanes, laying hens, alfalfa, outdoor consumption, free-range

## Abstract

**Simple Summary:**

Near-infrared spectroscopy (NIRS) has emerged as an accurate and promising alternative to traditional wet chemistry methods in feed and food science. Its applicability extends to estimating concentrations of compounds such as n-alkanes, based on their chemical properties, in various materials, including feed and feces. Analysis of excreta n-alkane patterns can provide insight into the dietary behavior of laying hens, particularly in scenarios where they have access to free-range areas and potentially consume plants from outside sources. Our study attempts to explore extreme cases, such as hens consuming only commercial feed, which result in lower concentrations of n-alkanes in excreta, thus challenging NIRS as a replacement for wet chemistry. Evaluating the accuracy of NIRS in predicting n-alkanes in excreta is critical because of its potential for substantial time and cost savings. Furthermore, it contributes to a deeper understanding of future nutritional strategies for laying hens, particularly in light of external nutritional contributions.

**Abstract:**

N-alkanes offer a promising approach for assessing the nutritional contribution of external sources to the diets of laying hens in free-range production systems. However, traditional laboratory methods, involving extraction, purification and gas chromatographic analysis, are both economically burdensome and time-consuming. Near-infrared spectroscopy (NIRS) is emerging as a viable alternative, with varying degrees of accuracy depending on the chemical nature and concentration of the component of interest. In our research, we focus on the accuracy of NIRS in predicting the concentrations of n-alkanes (C25–C33) in excreta under simulated free-range conditions with two different diets: one containing a commercial feed with minimal n-alkane content and another containing 1% alfalfa on top of the commercial feed. Spectra processing and calibration were tailored for each n-alkane, with NIRS performance influenced by diet type. Notably, plant predictions using NIR-generated data were consistent with laboratory results, despite a slight tendency toward overestimation (3.40% using the NIRS-generated C25-C29-C33 combination versus 2.80% using laboratory analysis). This indicates the potential of NIRS as an efficient tool to assess n-alkanes in excreta of laying hens and, consequently, the nutritional contribution of the free-range environment, providing rapid and cost-effective results.

## 1. Introduction

Historically, markers have played a fundamental role in monogastric animal studies, focusing primarily on digesta kinetics [1]. In this time, radioisotopes have emerged as the markers of choice [2,3,4]. External markers for poultry, whether added to the diet or administered directly to the animals, have been integral to poultry studies, including chromic oxide [5], barium sulfate [6], and titanium dioxide [7]. Challenges associated with external markers, such as poor mixing with certain feeds and legislative restrictions due to toxicity concerns, have prompted in recent years research into alternative approaches.

Long-chain normal alkanes (n-alkanes), hydrocarbons with predominantly odd carbon chains (C21-C37), have served as prominent plant biomarkers for nearly a century [8,9]. Widely used in ruminant studies under extensive grazing conditions, n-alkanes provide a consistent analytical procedure to be quantified in both forage and fecal samples, facilitating the assessment of botanical [10] and feed intake [11,12,13,14,15,16]. While extensively used in ruminants, there has been limited application of this methodology in monogastric animals, particularly poultry [17], which may be related to the poor content of n-alkanes in concentrated feeds. In a previous publication [18], we demonstrated the efficacy of n-alkanes found in diet components and excreta as markers to accurately predict the diet composition of individual laying hens, despite an overestimation of 2.80%.

The conventional methodology for quantifying n-alkanes involves a labor-intensive process that includes extraction, purification, derivatization, and gas chromatography-based quantitative analysis [19]. Given the associated costs and time requirements, more efficient methods for assessing n-alkane concentrations in plant species and feces are needed. Near-infrared (NIR) spectroscopy is emerging as a promising alternative, offering rapid, non-invasive and non-destructive analysis over a broad wavelength range (700 to 2500 nm) [20,21,22]. Through calibration and validation using extensive laboratory data and spectral information, NIR spectroscopy provides a rapid and efficient means of assessing n-alkane concentrations. Previous studies have demonstrated the effectiveness of NIR technology as a compelling replacement for wet chemistry in accurately predicting n-alkanes in fecal samples from various animal species, including cattle [22,23], sheep [24], deer [20], and horses [22].

The current research aimed to investigate the efficacy and accuracy of using NIR spectroscopy to estimate n-alkane concentrations in excreta samples from laying hens fed diets containing 1% of Alfalfa (*Medicago sativa*) as a simulation of a free-range scenario, and to predict the dietary composition of the hens using NIR generated data. We hypothesized that NIRS is a robust alternative for predicting n-alkanes in laying hens’ excreta samples. Any potential errors in prediction are expected to affect the accuracy of recovery rate estimates, thereby affecting the estimation of dietary composition through the associated equation. The accuracy of NIRS in this context is critical as it directly affects the reliability of the derived dietary composition, underscoring the importance of minimizing prediction errors for robust research outcomes.

## 2. Materials and Methods

### 2.1. Samples and Chemical Analysis

#### 2.1.1. Samples Collection

In total, 48 TETRA-SL laying hen pullets aged 18 weeks were housed in individual cages (ethics committee approval: MÁB-3/2023) and subjected to two dietary conditions. The control diet consisted of the standard recommendations outlined in the Tetra-SL guidelines, while the experimental diet included a 1% alfalfa supplement to the control diet. Over a 5-day adaptation period, daily individual feed intake and total excreta output were meticulously collected for each hen over 2 consecutive days. The collected samples were then carefully processed, including homogenization, weighing, lyophilization, and grinding, in preparation for the n-alkane analysis. Details on the animal experiment have been previously published [18].

#### 2.1.2. n-Alkane Analysis

The determination of n-alkane concentration followed a modified version of [25] and used gas chromatography as described by [18]. Briefly, 1 g of dried and finely ground samples (1 mm) were accurately weighed in duplicate, treated with 10 mL of 1.5 M KOH, and dosed with 0.15 g of n-tetracosane (C24) and n-tetratriacontane (C34) in undecane as an internal standard. The tubes were heated for 4.5 h at 90 °C in a water bath with continuous stirring, followed by partial cooling to 70 °C adding ice to the water bath. Subsequently, 8 mL of n-heptane and 5 mL of distilled water were added to each tube and shaken, and the process was repeated with 5 mL of heptane. The resulting extract was evaporated to dryness under a nitrogen blower to prevent oxidation, redissolved in 2 mL of heptane and gently applied to an SPE column (Discovery^®^ DSC-Si SPE Tube [500 mg, 3 mL] from Supelco Inc., Bellefonte, PA, USA) containing silica gel. The column was washed with 8 mL of heptane in four 2 mL steps. The eluent was evaporated to dryness, redissolved in 1 mL of heptane, and then transferred to the capillary column of a gas chromatograph for analysis. An Agilent 78902A (Agilent Technologies, Santa Clara, CA, USA) gas chromatography module, equipped with an Agilent 7693B automatic liquid sampler and a multimode inlet (split/splitless), was used for separating n-alkanes, together with a capillary column (Restek Rtx-1 30 m, 0.53 mm ID, 0.25 µm film thickness, with an upstreamed 5 m Integra-Guard column). The GC parameters were set as follows: injector temperature—300 °C; gas pressure—1.4774 psi; column—41.689 mL/min gas flow, 1 µL injection volume, splitless; oven temperature program—start at 100 °C, hold for 1 min, increase by 8 °C per minute to 310 °C, hold for 12.75 min; total run time—40 min; FI-detector settings—temperature of 320 °C, H_2_ flow of 35 mL/min, air flow (synthetic air) of 400 mL/min, makeup flow (N_2_) of 15 mL/min. The major (odd-chain) n-alkanes analyzed included C25, C27, C29, C31, and C33, with concentrations expressed in mg/kg DM.

### 2.2. Chemometric Analysis

A comprehensive dataset of laboratory analyses of specific n-alkanes (C25, C27, C29, C31, and C33) in excreta samples was carefully assembled and used to calibrate the Near Infrared Spectroscopy (NIRS) model in conjunction with the acquired spectra of the corresponding excreta samples.

#### 2.2.1. Excreta NIRS Acquisition

The NIR reflectance spectra of the 48 excreta samples were acquired using a PerkinElmer Frontier FT-NIR Spectrometer equipped with the accessory NIRA II (PerkinElmer, Shelton, CT, USA). The scanning process involved placing approximately 150 mg of each sample in a small ring cup that was previously cleaned with ethyl alcohol. Spectra were recorded over the wavelength range of 4000 to 10,000 cm^−1^ at 2 cm ^−1^ intervals with a resolution of 16 cm^−1^. Each spectrum was acquired over 120 scans to ensure robust and accurate spectral data.

#### 2.2.2. Spectra Processing

A crucial step in optimizing the accuracy and predictive capacity of models while minimizing complexity is the preselection of wavelengths during spectral processing, as pointed out by [26]. In the current research, the targeted spectral interval ranged from 4000 to 7500 cm^−1^, excluding the 7500–10,000 cm^−1^ interval based on the initial inflection point, as it did not provide a meaningful contribution to the calibration results. Various processing methods, including scatter correction methods such as multiplicative scatter correction (MSC) and standard normal variate (SNV), as well as mathematical treatments involving second derivatives at different smoothing points (ranging from 3 to 21), were systematically evaluated. Furthermore, potential combinations of the second derivative with MSC or SNV, considering all specified smoothing points, were explored to determine the most effective approach for spectral enhancement. Processing of the NIR spectral data was performed using the prospectr package [27] of the R program system.

#### 2.2.3. Calibration Development

Calibration involves developing a regression equation in which the reflected energy (log 1/R) of the NIR spectra are used as independent variables to predict the n-alkanes analyzed in the laboratory (dependent variable or reference). To ensure representative variability within both the calibration and validation data sets, the data were strategically divided into four intervals, each containing 25% of the observations. One observation from each group was randomly selected to contribute to the validation set in both the control and alfalfa groups. Hence, 40 excreta samples were allocated for calibration, while the remaining 8 were allocated for validation. Several processing techniques were used to build calibration models employing multivariate partial least squares (PLS) regression. Model calibration and validation were performed using the pls package [28] of the R program system. The accuracy of these models was assessed based on statistical metrics that explain the relationship between predicted and reference values, emphasizing the optimization of root mean square error of prediction (RMSEP) and coefficient of determination (R^2^) values while considering the minimum number of components. The processing combination that resulted in the lowest RMSEP was chosen.

#### 2.2.4. Model Validation

The assessment of the most suitable prediction models involved a rigorous cross-validation (CV) approach, employing a leave-one-out strategy. This repetitive process ensures that each sample is systematically excluded at least once from the calibration set, allowing the determination of the optimal model for a given dataset, as outlined by [29]. The resulting models were further checked for quality through external validation, where the remaining one-sixth (1/6) of the initial set considered as a validation set, and previously excluded during the calibration, served as an external sample to thoroughly evaluate the pre-selected optimal calibration models. Prediction accuracy was evaluated by estimating and comparing the root mean square error of calibration (RMSEC) and cross-validation (RMSECV) values, providing insight into potential overfitting, completed with the root mean square error of validation (RMSEV). Additionally, coefficients of determination for calibration (R^2^c), cross-validation (R^2^cv) and validation (R^2^v) were used to comprehensively assess model performance [30].

### 2.3. Calculations and Statistical Analyses

The accuracy of the data generated by the selected calibration model for each n-alkane was evaluated by analyzing the discrepancy of the committed error between the NIR-generated data and the one obtained from the laboratory analysis. To visually represent this comparison, Bland–Altman plots [31] were performed, which provide a comprehensive representation of the agreement between the predicted values from the calibration model and the actual laboratory measurements. This approach not only quantifies the accuracy of the model but also provides insight into potential systematic biases or trends in the data, ensuring a thorough assessment of the performance of the model across different n-alkanes. Recovery rates for each n-alkane were individually computed for each animal, considering data from both laboratory analysis and NIR-generated predictions separately. Adjustments to n-alkane concentrations in excreta were performed meticulously, considering the specific recovery rates associated with each diet based on predicted values obtained via NIRS. The approach suggested by [9] was implemented to estimate the proportion of alfalfa in the total feed intake. This involved using a non-negative least squares algorithm to solve a linear equation defined as x_f_F_i_ + x_a_A_i_ = E_i_, where F_i_, A_i_, and E_i_ (for i = 25, 27, 29, 31, 33) represent the concentrations of n-alkane i in the feed, alfalfa, and excreta, respectively, as reported by [18]. Once the coefficients x_a_ and x_f_ were determined, the proportion of dietary dry matter (DM) supplied by plants was calculated using the ratio x_a_/(x_a_ + x_a_). The nnls [32] package, an implementation of the non-negative least squares algorithm of the R program system, was used for this purpose. The predictive accuracy of the method was evaluated by calculating the mean square error (MSE) for predicting animal consumption within each diet, with further decomposition into variance and squared bias components. In addition, the proposed equation was systematically solved including all five n-alkanes studied and explored through all best combinations from two to five reported by [18]. This comprehensive approach ensures the selection of the most accurate and reliable n-alkane combination, providing confidence in the predictive accuracy of the method. All figures were generated using the ggplot2 package [33] of the R program system.

## 3. Results

Descriptive statistics for the selected datasets, which include both calibration and validation of the NIR model for each n-alkane individually, are shown in Table 1. The strategic structuring of the overall data set into four groups, with a random selection of one sample from each group for each treatment, facilitated the generation of calibration and validation data sets that exhibited consistent ranges for each component. This consistency is demonstrated by tightly matched means and standard deviations between the two data sets. Such consistency highlights the robustness and reliability of both the calibration and validation processes.

Figure 1 provides a visual representation of the raw data obtained from scanning the 48 excreta samples with NIRS. Notably, the first inflection point in the spectrum is observed at approximately the 7500 cm^−1^ wavelength. This observed inflection point prompted thorough calibrations, both inclusive and exclusive of the 10,000–7500 cm^−1^ range, in order to assess the impact of this highlighted area on the improvement of the results. The scattering effect between the different spectra of each sample is graphically evident, highlighting the need for scatter correction processing during the calibration phase.

Table 2 summarizes the optimal preprocessing approaches chosen for each n-alkane, which include detailed mathematical treatments with various smoothing points and required scatter corrections. The selection criteria for these methods were based on comprehensive statistical parameters, specifically RMSE and R^2^, evaluated for both calibration and cross-validation sets. These selected models also prioritized efficiency by incorporating the minimum number of components derived from the developed partial least squares regression model for each n-alkane.

The variation in smoothing points across different n-alkanes was considerable, ranging from 3 to 21, underscoring the need for adaptability based on the specific characteristics of each compound. Notably, the scatter correction processing showed distinct suitability for the n-alkanes studied, with SNV proving to be an appropriate choice for most, except for C29, where MSC was found to be more effective. A consistent minimum of three components was found to be optimal for all n-alkanes; however, for C29 a slightly more complex five-component model was deemed necessary to capture the nuances of its spectral characteristics.

In the calibration phase, the RMSE values for C25, C27 and C31 varied around 0.1, showing remarkable precision. However, a slightly higher RMSE of 0.54 and 0.81 was observed for C29 and C33, respectively. The coefficient of determination (R^2^) was above 0.80 for all n-alkanes, reaching 0.83 for C25, 0.81 for both C27 and C33, 0.92 for C31 and 0.94 for C29, underlining the robustness of the calibration models.

During the subsequent cross-validation, an increase in RMSE values was observed, with the highest values assigned to C31 at approximately 1.59 and C29 at around 1.19, indicating some variability. Conversely, C25, C27, and C33 maintained relatively low RMSE values of about 0.2. Notably, the R^2^ values showed a decrease for all investigated n-alkanes in the cross-validation round, particularly for C25, which recorded the lowest value among all n-alkanes around 0.22.

The external validation, performed with the eight randomly selected samples forming the validation set, demonstrated an improvement in RMSE and R^2^v for all n-alkanes except C33. In the case of C33, a decrease in R^2^v (0.55) and a slight increase in RMSE (0.261) were observed compared to the cross-validation metrics.

Figure 2 illustrates the selected calibration models of the investigated n-alkanes, providing key insights. It is notable that for C25, C27, and C33, the dispersion of the samples suggests a cohesive grouping, implying a consistent response pattern. In contrast, the grouping among animals receiving the two different diets (with or without alfalfa supplementation) is particularly pronounced for C29 and C31. The observed splitting in these cases implies a discernible influence of dietary variation, highlighting the sensitivity of the calibration models to specific dietary composition.

The assessment of NIR accuracy in predicting excreta n-alkane concentration, as shown in Table 3, includes a detailed breakdown of the errors committed, including bias, slope, and residual error.

Figure 3, derived from Bland–Altman plots, visually illustrates the dispersion of the validation set along the axes of predicted values versus differences.

A significantly higher sum of squared errors of prediction (SSEP) values of 6.92 and 9.35 were observed for C29 and C31, respectively, in contrast to the other n-alkanes studied, where SSEP values were below 0.45. In particular, C33 showed a unique pattern, with the slope contributing to 45% of the explained error, unlike C25 and C27, where the error committed relied mainly on the residual, which exceeded 90%. Furthermore, it is crucial to emphasize that the error resulting from the slope was relatively high for both C29 and C31 compared to C25 and C27, with values of (11.9 and 26.6) versus (2.83 and 2.21), respectively. This nuanced analysis sheds light on the various sources and magnitudes of error in the NIR predictions.

Based on the NIR-generated concentrations of the alkanes studied, individual recovery rates were accurately estimated, as shown in Table 4, along with values derived from laboratory analysis data. The calculated recovery rates based on NIR-generated data ranged from 32.8 to 44.3 for the commercial diet and from 36.7 to 47.6 for the mixed diet. These results were compared with the recovery rates calculated using laboratory analysis data, which ranged from 29.7 to 43.9 for the commercial diet and 37.2 to 47.9 for the diet. The application of NIR as a predictive tool revealed notable differences in the estimated recovery rates. The decrease in the recovery of n-alkanes observed in animals consuming the diet containing 1% alfalfa was particularly noticeable, with the major change observed for C31, which showed a decrease of 3.89% compared to the values estimated based on laboratory data. Conversely, a remarkable increase was observed in the estimates derived from animals consuming the commercial diet. In particular, C29 and C31 showed an increase of 8.61% and 16.5%, respectively, compared to the reference data obtained from laboratory analysis.

Figure 4 presents the estimate of alfalfa as a percentage of total feed intake per animal on a dry matter basis, derived from the solution of the linear equation involving all studied n-alkanes. Data are grouped according to the diet received, with significantly higher estimates observed for the commercial diet. The highest estimation, reaching approximately 0.37% of the alfalfa inclusion, was observed for animal 20. In contrast, the mixed diet had the highest estimate for animal 24, with a total alfalfa inclusion of 1.31%.

Various statistical metrics, including RMSE, variance, and bias, were used to evaluate the accuracy of these estimates and summarized in Table 5. For animals on the commercial diet, the average alfalfa inclusion rate varied around 0.017 and 0.066 using laboratory and NIR data, respectively. Estimation using NIR predictions showed a relatively higher RMSE of approximately 0.126 compared to 0.032 using laboratory data. Conversely, for animals receiving a diet containing 1% alfalfa, the precision remained consistent, registering around 0.80% for both laboratory and NIR-generated data, indicating a moderate underestimation of 0.20. It is noteworthy that the use of NIR-generated n-alkane concentration data for all n-alkanes studied demonstrated a relatively accurate prediction of the proportion of alfalfa added to the diet on a dry matter basis, depending on the specific diet received.

The identified optimal combinations of n-alkanes for predicting the proportion of alfalfa in total dry matter intake (%) in hens, based on the laboratory analysis reported by [18], were subjected to testing and evaluation using NIRS-generated data, as illustrated in Table 6. The combination C25-C27-C29-C31-C33 including all studied n-alkanes, resulted in an estimate of 0.808%, demonstrating a significant underestimation of 19.2%. In contrast, the combination C25-C27-C29-C33 showed a more modest underestimation of 2.8%, resulting in an average alfalfa incorporation value of approximately 0.972%. However, the reported most effective combination of all the possibilities identified as C25-C29-C33, showed a slight overestimation of 3.4%, resulting in a mean alfalfa incorporation value of approximately 1.034%. In addition, C25-C29 resulted in an overestimation of approximately 4.4%.

It is noteworthy that the overestimation of n-alkane concentrations using NIRS had a marginal effect on the accuracy of alfal-fa incorporation, especially for the best combination defined as C25-C29-C33. An increase in bias of about 21% (0.034/0.028) was observed when compared to estimates using laboratory data. This emphasizes the need for careful consideration of the selected n-alkane combinations and the potential influence of NIRS-generated data on the accuracy of alfalfa proportion predictions.

## 4. Discussion

Near-infrared (NIR) spectroscopy stands as a highly valuable technology for laboratory analysis across diverse chemical components. Its notable capabilities include enabling simultaneous qualitative and precise quantitative analysis of multiple parameters, facilitating high sample throughput, and providing real-time monitoring capabilities [34].

In the current research, the adoption of the second derivative as the appropriate mathematical treatment for data derived from laying hen excreta samples, coupled with SNV as the primary scatter correction processing, underscores the complex nature of spectroscopic analyses. This choice stands in contrast to the findings of [22], who identified the first derivative as optimal for calibrations using bovine and equine fecal samples, particularly in the 4000–9000 cm^−1^ range.

The higher RMSE observed during cross and external validations for C29 and C31 could be attributed to the significant contribution of the 1% alfalfa to these compounds, even at a low inclusion rate. These n-alkanes are known to be present at high concentrations in both plants [35] and fecal samples [36], which is supported by various studies [15,18,22,37,38,39]. The precision of the calibration was also affected by the grouping of the samples in the PLS regression, especially for the interval between 3 and 6 mg/kg DM as shown in Figure 2, which provides a lack of information regarding the trend of the regression for this interval. An experimental design covering this range is required to enhance the accuracy, especially for the n-alkanes C29 and C31.

The cross-validation coefficient of determination (R^2^cv) showed commendable values, around 0.70, for C29, C31 and C33 in our study. These results are consistent with the conclusions of [22], who reported R^2^cv values above 0.71, with a tendency to be higher for odd-chain n-alkanes with higher concentrations, especially C29 and C31. This agreement with previous studies, including those by [20,24] using deer and sheep fecal samples, respectively, underscores the robustness of NIR spectroscopy in predicting alkane concentrations. In contrast, the R^2^cv values of 0.22 and 0.48 for C25 and C27, respectively, appear relatively low compared to the wide range of 0.84 to 0.91 reported by [22], depending on the specific n-alkane and animal species. This discrepancy suggests potential differences in the predictability of specific alkane concentrations across studies.

An interesting observation relates to the prediction accuracy for C31 in bovine feces, where [22] reported an R^2^cv of 0.14 and a substantial RMSEP of 293, in contrast to our model, which showed moderately high accuracy with an R^2^cv of 0.69 and a RMSEP of 1.60. The possible explanation for this discrepancy lies in the non-linear structure of C31, which corresponds to a specific spectrum and may have a greater interference effect, as noted by [40]. Furthermore, notable differences in RMSEP were observed between the two studies. Ref. [22] reported significantly higher values ranging from 5.1 to 293, whereas the results of the present study revealed a narrower range of 0.16 to 1.60.

The observed differences in n-alkane concentrations derived from laboratory analysis versus NIR-generated predictions, particularly noticeable for C29 and C31 in the commercial diet, led to corresponding differences in recovery rates. Compared to the laboratory data reported by [18], where recovery rates ranged from 0.30 to 0.48, the NIR-generated data showed a slightly different range of 0.33 to 0.48.

The observed improvement in R^2^ during external validation in the current study provides confidence in the robustness of the model, indicating that it is not overfitting to the training data, but rather capturing authentic patterns in the underlying data. This upward trend in R^2^v serves as a positive indicator, suggesting that the predictive ability of the model effectively extends to new, unseen data. By contrast, the results reported by [22] tend to show a slight decrease in R^2^ during external validation for almost all the studied n-alkanes.

The larger contribution of the slope to the committed error in predicting n-alkanes C29, C31, and C33 underscores the sensitivity of the prediction accuracy to the concentration of these n-alkanes in the samples. The observed trend indicates that as the concentration increases, the committed error tends to be larger due to discernible differences in concentration between two groups of diets. Notably, the calibration model for C33 displays a consistently low committed error compared to C29 and C31, a finding attributed to the high concentration of C29 (289 mg/kg DM) and C31 (358 mg/kg DM) in alfalfa, as reported by [18].

To the best of our knowledge, this study represents the first exploration of NIR as a predictive tool for n-alkanes in excreta samples from laying hens. While n-alkanes are predominantly used for ruminants or wildlife [9,15,41,42], our results suggest a comparable trend in the applicability of NIR, reflecting the influence of species on predictive accuracy, in agreement with [22]. In the current study using excreta samples from laying hens, the R^2^cv ranged from 0.50 to 0.70, with C25 demonstrating a lower coefficient (0.22). Contrasting results were reported by [20,24], who reported higher R^2^cv (>0.80) for the same n-alkanes in fecal samples from fallow deer and sheep, respectively. Ref. [22] reported relatively lower R^2^cv values for cattle (0.19 for C31) and horse (0.29 for C33) feces. Notably, our calibration model selection criterion prioritized the lowest RMSE for both calibration and cross-validation, with the maximum value recorded being approximately 1.6 for all n-alkanes studied.

Interestingly, a notable overestimation of approximately 8.61% and 16.5% was observed for the recovery rates of n-alkanes C29 and C31, respectively, when using the NIR-generated data for the commercial diet. This discrepancy highlights the potential impact of predictive technologies on recovery rate estimates and underscores the need for careful validation and adjustment when transitioning from laboratory to NIR-based analyses.

Alfalfa proportion of total dry matter intake estimation, following the procedure of [9] and adopted by [18], showed consistent results using both laboratory and NIR-generated data for the mixed diet containing 1% alfalfa, resulting in a 20% overestimation using the five n-alkanes studied. Conversely, for the commercial diet, the NIR-generated data showed a significant increase, approximately 300% higher mean values (0.017/0.066) compared to the laboratory data reported by [18]. This difference was reflected in a higher RMSE, which increased from 0.032 for the laboratory data to 0.126 for the NIR predictions. The substantial variations were attributed to the increased contribution of alfalfa in the diet, especially for C29 and C31, where a 1% inclusion of alfalfa resulted in a 200% and 300% increase, respectively, as reported by [18]. Consequently, the concentrations of n-alkanes in the excreta were affected, resulting in a modification of the recovery rate. In the present research, which was designed to simulate real-life free-range scenarios, calibration equations were developed using data from both diets together, which resulted in a clear separation of the two groups, particularly noticeable in Figure 2 for n-alkanes C29 and C31. However, this separation resulted in a lack of information about the trend of the regression between the two diet groups, and consequently affected the precision of n-alkanes C29 and C31.

The noted source of imprecision in estimating n-alkanes, particularly C29 and C31, affected the accuracy of predictions using the optimal n-alkane combinations proposed by [18]. This resulted in an increase in the RMSE, while the mean values of alfalfa percentage of total dry matter intake appeared to be slightly overestimated. The C25-C29-C33 combination identified as the best solution by [18] showed an overestimation of 2.80% compared to the target of 1% in the experiment. Nevertheless, the final estimate of 1.034%, reflecting an overestimation of 3.4%, is considered highly accurate, underscoring the robust predictive capabilities of the selected n-alkane combination.

It is important to acknowledge that the calibration and selection of n-alkane combinations have been mainly developed for alfalfa, which might limit their applicability to other plant species. Therefore, more comprehensive and robust calibrations are needed to accurately predict the dietary composition using excreta from animals consuming a mixture of plants.

## 5. Conclusions

In conclusion, NIR emerges as a viable alternative to laboratory analysis for predicting n-alkane concentrations in excreta samples from laying hens. Beyond the inherent committed error, mainly due to unexplained residuals, the data generated from these calibrations not only demonstrated precision but also proved effective in predicting quantitative dietary composition over the proposed procedure. It is crucial to emphasize the critical role of the careful selection of the combination of n-alkanes for the estimation of alfalfa fraction in the diet, with C25-C29-C33 identified as the most optimal combination for both laboratory and NIR-generated data. However, future studies should investigate the effect of increased doses of n-alkane source incorporation on the accuracy of the NIR technology in predicting their concentrations in excreta samples. In addition, more research should be conducted to improve the accuracy, especially for extremely low levels of incorporation. Such investigations, when applied to real-life free-range scenarios, will deepen our understanding of the contribution of free-range to a dietary plan. Moreover, it offers the advantage of saving time by providing immediate results and cost-effectiveness by avoiding the need for extensive laboratory analysis.

## Figures and Tables

**Figure 1 animals-14-00806-f001:**
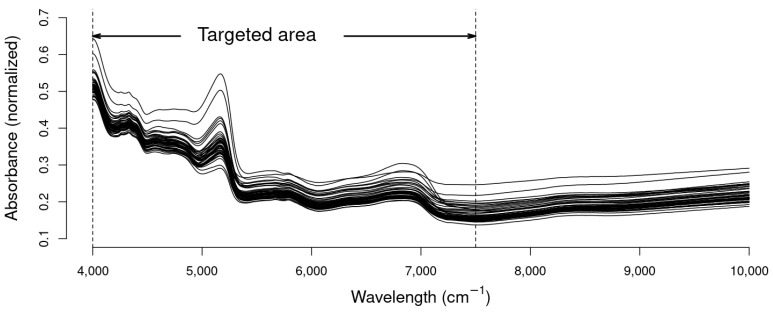
Near-infrared spectrum of laying hens excreta samples.

**Figure 2 animals-14-00806-f002:**
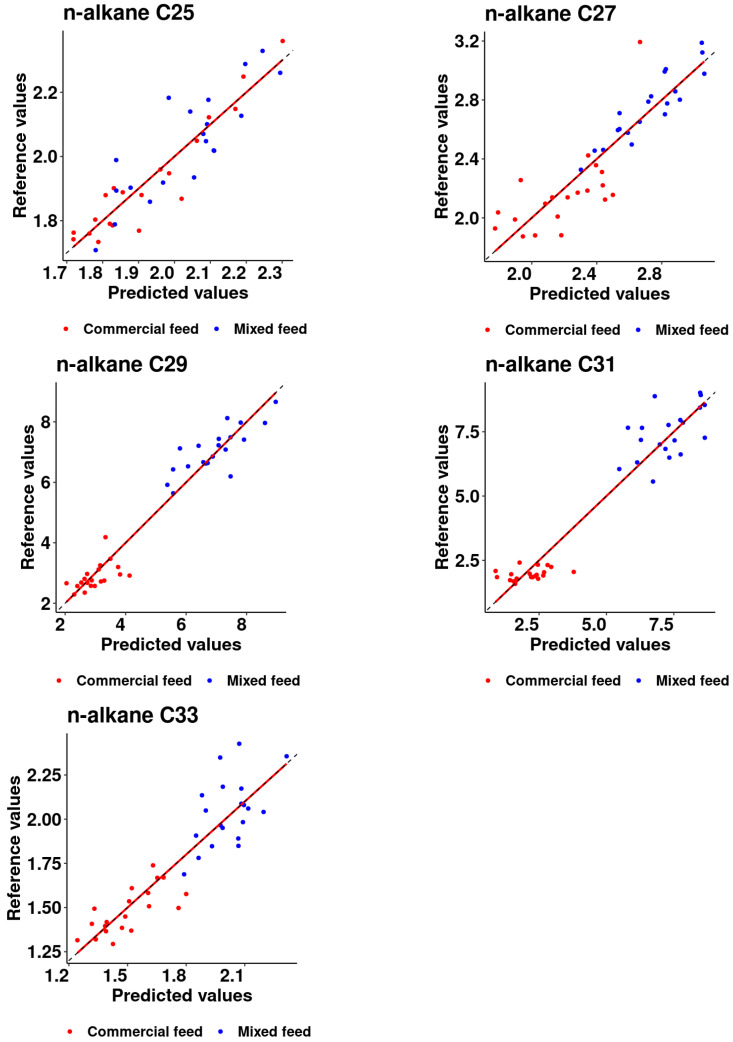
Scatter plot of reference versus predicted values (mg/kg DM) obtained with the calibrated model.

**Figure 3 animals-14-00806-f003:**
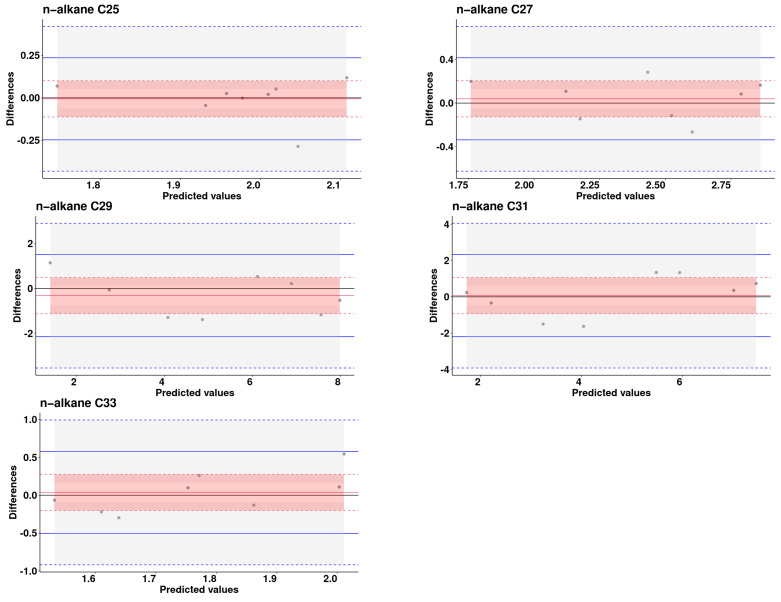
Bland–Altman plot: Predicted vs. differences analysis for n-alkanes concentrations in excreta samples from laying hens (Solid red line is the bias, the dashed red lines are the limits of the 95% confidence interval of the bias. Solid blue lines are the 95% limits of agreement, dashed blue lines are the 95% confidence interval limits for the limits of agreement).

**Figure 4 animals-14-00806-f004:**
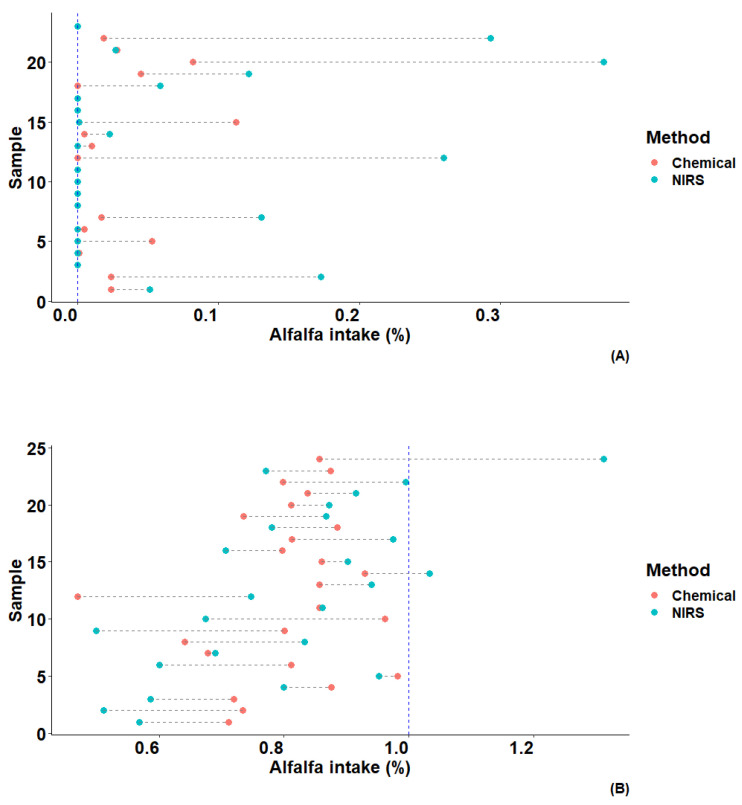
Predicted alfalfa proportion in total dry matter intake (%) of hens fed a commercial diet (**A**) and a diet with 1% alfalfa (**B**) using laboratory analyzed values of the studied n-alkanes.

**Table 1 animals-14-00806-t001:** Descriptive statistics of calibration and validation data sets of laying hens excreta n-alkanes (mg/kg DM).

n-Alkane	Calibration Set			Validation Set		
	n	Mean	SD ^1^	n	Mean	SD
C25	40	1.98	0.18	8	1.97	0.15
C27		2.46	0.39		2.45	0.39
C29		4.97	2.21		4.88	2.19
C31		4.71	2.87		4.71	2.93
C33		1.76	0.33		1.81	0.41

^1^ Standard deviation.

**Table 2 animals-14-00806-t002:** Calibration and cross-validation statistics of the excreta NIRS equations to predict the n-alkanes.

**n-Alkane**	**Math Treatment ^1^**	**Scatter Correction ^2^**	**Components Numbers**	Calibration		Cross-Validation		External Validation	
				RMSEC ^3^	R^2^c ^6^	RMSECV ^4^	R^2^cv ^7^	RMSEV ^5^	R^2^v ^8^
C25	2,13,1	SNV	3	0.074	0.83	0.158	0.22	0.115	0.31
C27	2,15,1	SNV	3	0.169	0.81	0.278	0.48	0.184	0.74
C29	2,21,1	MSC	5	0.536	0.94	1.186	0.70	0.930	0.79
C31	2,13,1	SNV	3	0.807	0.92	1.589	0.69	1.081	0.84
C33	2,3,1	SNV	3	0.141	0.81	0.205	0.60	0.261	0.55

^1^ The digit represents, respectively, derivative order, first smoothing, and second smoothing. ^2^ SNV = standard normal variate; MSC = multiple scatter correction. ^3^ RMSEC = root mean square error of calibration. ^4^ RMSECV = root mean square error of cross-validation. ^5^ RMSEV = root mean square error of external validation. ^6^ R^2^c = calibration coefficient of determination. ^7^ R^2^cv = cross-validation coefficient of determination. ^8^ R^2^v = external validation coefficient of determination.

**Table 3 animals-14-00806-t003:** Evaluation of predicted n-alkanes concentrations in excreta via NIRS vs. laboratory-analyzed values with committed error breakdown.

	SSEP ^1^	Bias (%) ^2^	Slope (%) ^3^	Residual (%) ^4^
C25	0.106	0.00	2.83	97.2
C27	0.271	4.43	2.21	93.4
C29	6.920	11.8	11.9	76.3
C31	9.346	0.39	26.6	73.0
C33	0.445	2.02	44.9	53.0

^1^ Sum of squared errors of prediction for each equation. ^2^ % of the SSEP is due to the bias. ^3^ % of SSEP is due that the slope is different from 0. ^4^ % of SSEP is due to random variation.

**Table 4 animals-14-00806-t004:** Estimated means of excreta recovery using laboratory analysis and predicted values via NIRS.

Diet n-Alkanes	Commercial Feed					Mixed Feed				
	C25	C27	C29	C31	C33	C25	C27	C29	C31	C33
Lab estimated mean *	0.439	0.369	0.302	0.297	0.339	0.479	0.375	0.372	0.437	0.409
NIR estimated mean	0.443	0.375	0.328	0.346	0.349	0.476	0.370	0.367	0.420	0.399

* Data published by [18].

**Table 5 animals-14-00806-t005:** Prediction of the proportion of alfalfa in total dry matter intake (%) in hens fed a diet with 1% alfalfa using data of all the studied n-alkanes coming from laboratory analysis vs. NIRS generated.

		Mean ^1^	RMSE ^2^	Bias	Var ^3^
Laboratory Data *	Commercial diet	0.017	0.032	0.017	0.001
	Mixed feed diet	0.804	0.224	−0.196	0.012
NIRS Data	Commercial diet	0.066	0.126	0.066	0.011
	Mixed feed diet	0.808	0.266	−0.192	0.034

* Data published by [18]. ^1^ Mean of prediction (n = 24). ^2^ RMSE root mean squared error of prediction. ^3^ Var variance of the error of prediction.

**Table 6 animals-14-00806-t006:** Test of the best possible combinations of n-alkanes to predict the proportion of alfalfa in total dry matter intake (%) in hens fed a diet with 1% alfalfa using data from laboratory analysis vs. NIRS generated.

		Mean ^1^	RMSE ^2^	Bias	Var ^3^
Laboratory Data *	C25-C27-C29-C31-C33	0.804	0.224	−0.196	0.012
	C25-C27-C29-C33	0.965	0.133	−0.035	0.016
	C25-C29-C33	1.028	0.124	0.028	0.015
	C25-C29	1.044	0.150	0.044	0.020
NIRS Data	C25-C27-C29-C31-C33	0.808	0.266	−0.192	0.034
	C25-C27-C29-C33	0.972	0.239	−0.028	0.056
	C25-C29-C33	1.034	0.240	0.034	0.056
	C25-C29	1.044	0.214	0.044	0.044

* Data published by [18]. ^1^ Mean of prediction (n = 24). ^2^ RMSE root mean squared error of prediction. ^3^ Var variance of the error of prediction.

## Data Availability

The data presented in this study are available in article.

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
