# Peer review of "Near-Infrared (NIR) Spectroscopy as an Alternative for Predicting n-Alkane Concentration in Excreta of Laying Hens: NIR-Generated Data for Dietary Composition Estimation"

_animals, 2024, doi:10.3390/ani14050806_

Round 1

Reviewer 1 Report

Comments and Suggestions for Authors

Thanks to the authors for this well prepared manuscript and that they as well point out the limitations of their findings in the discussion. I have just some minor comments and suggestions.

Line 80-91  Briefly,…   can be deleted as well. The auhors already avoided to repete the experimental conditions. Therefore it would be logical to delete as well the short description of the n-alkane analysis using gas chromatography because it is only  repetition of the previous paper. Giving the reference would be sufficient.

Table 1 : What is the unit of these values…I presume mg/kg DM as I can read later somewhere please include. Maybe you can include somewhere in the Material and Method section that calculations are based on mg/kg DM

Is it possible to include as well the range of the data (lowest and highest) for each n-alkane?

Figure 1  Absortion ? Is that French?  Should be Absorbance .

Table 2 :These values (RMSE ; R-square) are not in mg/kg DM

Figure 2… What is the unit of reference and predicted values. Please include

Author Response

  • Detailed Methodology: We have included a comprehensive section detailing the methodology of the n-alkane analysis, considering other comments and considering it important as authors, as suggested (lines 91-104).
  • Clarification of Table 1 units: The unit for Table 1 has been updated to (mg/kg DM), and this adjustment has been reflected in both the table title (line 197) and the Materials and Methods section (line 105).
  • Avoidance of repetition: We acknowledge that the range of n-alkanes was previously published in Figure 2 (Dardabou et al., 2024). Therefore, to avoid redundancy, we have refrained from repeating these data.
  • Figure 1 Absorbance correction: The absorbance term in Figure 1 have been corrected according to the reviewer's instructions.
  • Units for RMSE and R-squared: The units (mg/kg DM) have been removed from the RMSE and R-squared values.
  • Units for reference and predicted values in Figure 2: The unit (mg/kg DM) has been added for both reference and predicted values in Figure 2.

Reviewer 2 Report

Comments and Suggestions for Authors

The exploration of Near-Infrared Spectroscopy (NIRS) as an alternative to traditional wet chemistry methods in feed and food science is both fascinating and promising. This manuscript acknowledges the increasing accuracy and reliability of NIRS, particularly in estimating concentrations of compounds like n-alkanes based on their chemical properties. The versatility of NIRS in analyzing various materials, including feed and feces, showcases its potential to revolutionize research methodologies.

The manuscript's focus on analyzing excreta n-alkane patterns in laying hens offers valuable insights into their dietary behavior, especially in scenarios where hens have access to free-range areas and may consume plants from external sources. The decision to explore extreme cases, such as hens consuming only commercial feed, adds depth to the study by challenging the efficacy of NIRS in comparison to wet chemistry methods. This nuanced approach contributes significantly to the ongoing conversation about the application of NIRS in feed and food science.

I'm not an expert in Near-Infrared Spectroscopy but this limitations does not limit the accessibility of the study to a broader audience, making it more inclusive for readers with varying levels of expertise.

Additionally, the well-designed experimental approach allows for meaningful correlations between NIRS estimation and alkane quantification by lab methodologies underscores the robustness of the study. The emphasis on evaluating the accuracy of NIRS in predicting n-alkanes in excreta is particularly noteworthy, considering the potential for significant time and cost savings in comparison to traditional wet chemistry methods.

Author Response

We would like to express our sincere gratitude for the time and expertise you devoted to reviewing our manuscript. Your insightful remarks and positive feedback were invaluable to us.

Your encouragement and recognition of the strengths of our research are deeply appreciated. It is immensely gratifying to receive such positive feedback from a respected expert in the field.

Once again, thank you for your time, effort, and encouraging words. Your contribution is instrumental in shaping the final version of our manuscript.

Reviewer 3 Report

Comments and Suggestions for Authors

General comments

The manuscript Near-Infrared (NIR) Spectroscopy as an Alternative for Predicting n-Alkane Concentration in Excreta of Laying Hens: NIR-Generated Data for Dietary Composition Estimationproposed for publication in Animals aimed to investigate the efficacy and accuracy of using NIR spectroscopy to estimate n-alkane concentrations in excreta samples from laying hens fed diets containing plant supplementation as a simulation of a free-range scenario. Moreover, a prediction of the dietary composition of the hens using NIR generated data was assessed.

Overall, the manuscript is well written and designed. Interesting and relevant topic is discussed. However, some corrections need to be made before being published. Below specific comments:

Specific comments

Introduction

- Please add a paragraph with studies regarding n-alkanes analysis using NIRS.

- L 54: provide any previous study regarding assessment of n-alkane concentrations with NIR spectroscopy.

- L 58: “diets containing plant supplementation”. Please be specific. Which plant? What supplemented level?

Materials and Methods

- L 77: “… have been published elsewhere” replace with “have been previously published”.

- L 88: “SPE column containing silica gel”. Add specific details for SPE column – model, producer.

- L 90: mention the specific details for chromatographic column (dimensions, model, producer…), and also for GC equipment.

- the GC method for n-alkanes analysis is not presented. Please add it!

- The GC method is considered the reference method for n-alkanes analysis?

The Results are well organized and clearly presented. The Discussion is concise, there are presented also limitations of the study’ applicability, and future perspectives.

The Conclusions are justified by the manuscript’ results.

The References should be updated with more recent articles.

Author Response

  • NIRS analysis background: Additional information and references regarding NIRS analysis of n-alkanes have been incorporated into the paragraph spanning lines 55 to 57, supplemented by relevant references introduced in line 54.
  • Description of plant species and supplementation levels: Details on plant species and level of supplementation were added to line 61 to provide a more complete understanding of the study.
  • Clarity in previously published work: "have been published elsewhere" has been replaced with "have been previously published" for clarity and consistency in lines 79-80.
  • Methodological clarification: Specific details about the SPE column (line 91) and chromatographic column (line 98), as well as the GC methodology (lines 95-105) have been added to improve methodological clarity and completeness.